# The association between the gut microbiome and 24-h blood pressure measurements in the SCAPIS study
Yi-Ting Lin [1,2], Sergi Sayols-Baixeras [1,3], Gabriel Baldanzi [1], Koen F. Dekkers [1], Ulf Hammar[1], Diem Nguyen [1], Nynne Nielsen [4], Aron C. Eklund [4], Georgios Varotsis [1], Jacob B. Holm [4], H. Bjørn Nielsen [4], Lars Lind[5], Göran Bergström [6,7], J. Gustav Smith[8,9,10], Gunnar Engström[11], Johan Ärnlöv[12,13], Johan Sundström [5,14], Marju Orho-Melander [11] & Tove Fall [1] ✉

## Abstract

**Background** There is mounting evidence supporting the role of the microbiota in hypertension from experimental studies and population-based studies. We aimed to investigate the relationship between specific characteristics of the gut microbiome and 24-h ambulatory blood pressure measurements.

**Methods** The association of gut microbial species and microbial functions, determined by shotgun metagenomic sequencing of fecal samples, with 24-h ambulatory blood pressure measurements in 3695 participants and office blood pressure was assessed in multivariable-adjusted models in 2770 participants without antihypertensive medication from the Swedish CArdioPulmonary bioImage Study.

**Results** Gut microbiome alpha diversity was negatively associated with diastolic blood pressure variability. Additionally, four microbial species were associated with at least one of the 24-h blood pressure traits. *Streptococcus* sp001556435 was associated with higher systolic blood pressure, *Intestinimonas massiliensis* and *Dysosmobacter* sp001916835 with lower systolic blood pressure, *Dysosmobacter* sp001916835 with lower diastolic blood pressure, and ER4 sp900317525 with lower systolic blood pressure variability. Moreover, office blood pressure data from a subsample without ambulatory blood pressure measurements replicated the association of *Intestinimonas massiliensis* with systolic blood pressure and *Dysosmobacter* sp001916835 with diastolic blood pressure. Species associated with 24-h blood pressure were linked to a similar pattern of metabolites.

**Conclusions** In this large cross-sectional analysis, gut microbiome alpha diversity negatively associates with diastolic blood pressure variability, and four gut microbial species associate with 24-h blood pressure traits.

## Plain language summary

High blood pressure is a major risk factor for heart disease and stroke. Recent research suggests that bacteria in the gut may influence blood pressure, but more studies are needed. In this study, we analyzed data from 3695 people in Sweden who wore a 24-h blood pressure monitor and provided stool samples for gut microbiome analysis—a method that identifies the types and abundance of microorganisms in the sample. We found that people with greater gut bacteria diversity had more stable blood pressure. Additionally, four specific bacterial species were linked to blood pressure levels; three were associated with lower blood pressure, while one was linked to higher levels. These findings suggest that gut bacteria may play a role in regulating blood pressure. Future research could explore whether changing the gut microbiome might help prevent or treat high blood pressure.

Hypertension is estimated to have contributed to 10.8 million deaths globally in 2019 through its impact on cardiovascular and renal disease development[1]. Yet, our understanding of hypertension pathophysiology is incomplete, although it has been hypothesized that the gut microbiota affects the blood pressure (BP) regulation mediated through microbiota-derived metabolites and interactions with the immune system[2]. Evidence supporting the role of the microbiota in hypertension is based on experimental and observational studies. Germ-free mice transplanted with fecal

material from hypertensive human donors showed higher systolic (SBP) and diastolic BP (DBP) than mice transplanted with material from a normotensive donor[3]. Plasma metabolites derived or modified by the gut microbiota, including trimethylamine N-oxide (TMAO) and short-chain fatty acids (SCFA), have been suggested to affect blood pressure through chemosensory receptors[4–6].

In 4672 participants from the HEalthy Life In an Urban Setting (HELIUS) study, where the gut microbiota was characterized using the 16S

amplicon method, inverse associations of SCFA-producing bacteria with SBP were observed[7]. However, findings were inconsistent across ethnicities. Another large study based on shallow metagenomics sequencing in samples from 6953 participants of the FINRISK 2002 study identified mainly positive associations between 45 microbial genera and BP. Studies applying methods resulting in a higher resolution of the microbiome are anticipated to provide more specific results.

Moreover, while SBP and DBP measured at the test centers constitute a snapshot of BP, ambulatory blood pressure monitoring (ABPM) provides a more representative measurement of SBP and DBP. ABPM is not affected by white-coat hypertension, and it captures nocturnal BP. ABPM can also be used to estimate the variability of BP, often defined as the standard deviation (SD) of SBP and DBP, and has been reported as an independent risk factor for cardiovascular disease[8]. However, there is a lack of population-based studies investigating the association between the gut microbiota and BP using ABPM for capturing BP and BP variability.

In this study of participants without antihypertensive medication, we find that gut microbiome alpha diversity is negatively associated with DBP variability and identify four microbial species associated with 24-h BP traits. Specifically, *Streptococcus* sp001556435 associates with higher SBP, while *Intestinimonas massiliensis* and *Dysosmobacter* sp001916835 associate with lower SBP, and *Dysosmobacter* sp001916835 also with lower DBP. Additionally, ER4 sp900317525 is linked to reduced SBP variability. These associations are replicated in office BP measurements and are supported by consistent metabolic profiles, suggesting a potential role of specific gut microbiota in blood pressure regulation.

## Methods

### Study population

The Swedish CArdioPulmonary bioImage Study (SCAPIS) cohort is a prospective observational study with the baseline investigation conducted in 2013–2018, including 30,154 individuals invited from a random extract from the population register of individuals aged 50–64 living in 6 counties across Sweden, including 2–3 visits to the respective study centers[9]. We used data from the baseline investigation from the Uppsala ($n = 4839$) and Malmö ($n = 4977$) centers with available fecal metagenomics data (Supplementary Fig. S1). We excluded 2458 participants who had a prescription for antihypertensive medication in the last 12 months in the Swedish Prescribed Drug Register (ATC codes in Supplementary Methods) before the first baseline visit and 893 participants with missing information on at least one covariate or missing blood pressure measurement. A subset of the remaining 6465 individuals ($n = 4007$) was provided with equipment for ABPM. However, data from 312 individuals did not pass the quality control[10], and thus, leaving 3695 individuals who formed the ABPM subsample with high-quality ABPM data, no antihypertensive medication, and complete data on covariates for the main analysis. The remaining 2770 participants with no antihypertensive medication and complete data on covariates and valid office BP measurements formed the non-ABPM subsample. All participants provided written informed consent. The study was conducted in accordance with the Declaration of Helsinki. The Swedish Ethical Review Authority approved the Swedish CardioPulmonary bioImage Study (DNR 2010-228-31M) and the present study (DNR 2018-315).

### Blood pressure measurements

Participants of the ABPM subsample were instructed to apply the ABPM device (Labtech EC-3H/ABP, Labtech Ltd, Debrecen, Hungary) in the morning and to remove it 24 h later. SBP and DBP were measured every 30 min during the day, and every 90 min during the night in participants from the Uppsala center[10]. SBP and DBP were measured automatically every 30 min for participants from the Malmö center. Office BP was measured after five minutes of rest in supine position using an automatic device at the brachial artery (Omron M10-IT, Omron Healthcare Co. Ltd, Kyoto, Japan). Details are provided in Supplementary Methods.

### Fecal sample processing and metagenomic sequencing

Fecal samples were collected for microbiome analysis[11]. At the first visit, participants received a fecal sample collection kit containing barcoded tubes, gloves, re-sealable plastic bags, paper collection bowls, and instructions from the SCAPIS study center. Participants were instructed to collect fecal samples at home and to store the tubes in sealed plastic bags in their home freezer until the second visit to the study center, where they were then stored for a maximum of 7 days at −20 °C until they were transferred to the −80 °C freezers at the central biobank. The samples were shipped on dry ice to Cmbio A/S (Copenhagen, Denmark), where DNA extraction, metagenomic shotgun sequencing, and bioinformatics analyses were performed[11]. DNA extraction was performed utilizing NucleoSpin 96 Soil kits, with each extraction round incorporating both negative and positive controls to ensure methodological rigor. Following DNA fragmentation and library preparation, shotgun metagenomic sequencing was conducted using the Illumina Novaseq 6000 system (Illumina, USA), yielding an average sequence depth of 25.3 million read pairs for the Uppsala samples and 26.3 million read pairs for Malmö samples. Metagenomic sequence data were analyzed using the Clinical Microbiomics Human Microbiome Profiler (CHAMP) with the Clinical Microbiomics Human Microbiome Reference (HMR05) gene catalog[12], resulting in species-level relative abundance estimates of eukaryotes and prokaryotes. Prokaryotic species were annotated using the Genome Taxonomy Database (GTDB) release 214. A total of 4594 microbial species were identified. Of these, 268 species met our inclusion criteria of having a relative abundance >0.01% in more than 30% of study participants. We applied centered-log ratio (CLR) transformation on the species relative abundances after an addition of a universal pseudo-value equal to the lowest non-zero abundance. After the CLR transformation, those values that were originally zero were replaced with the minimal non-zero transformed value for each species. This replacement ensured that all values that were equal to zero before the transformation would have an equally low value after the transformation. A rarefied species abundance table was generated through random sampling of 190,977 gene counts per sample. Microbial diversity was comprehensively evaluated using three alpha diversity metrics calculated from the rarefied abundance table. The functional potential profile of the gut microbiota was determined by assigning genes to 103 metabolic pathways comprising the gut metabolic modules (GMM)[13]. A species was considered to carry a GMM if it carried at least two-thirds of the KEGG (Kyoto Encyclopedia of Genes and Genomes) Orthology of a module. For modules with three or fewer steps, all steps were required. For modules with alternative paths, only one path had to fulfill the criterion.

De-identified de-hosted metagenomic sequencing data for SCAPIS samples can be accessed from the European Nucleotide Archive under accession number PRJEB51353, as detailed in Dekkers et al.[11].

### Plasma metabolites sample collection and processing

Blood specimens were obtained from study participants who fasted overnight prior to their clinical visit. All collected samples underwent plasma separation before being preserved at −80 °C within the facility's biorepository before being transported to the analytical laboratory at Metabolon Inc. (Durham, North Carolina) for comprehensive metabolomic evaluation. Details are given in the Supplementary Methods and in Dekkers et al.[11].

### Other phenotypes

Information on smoking, country of birth, previous diagnosis of diabetes and inflammatory bowel disease (Crohn's disease and ulcerative colitis), use of antidiabetic and antihyperlipidemic medications, and diet was based on questionnaire information. Smoking was categorized as either current smoker or non-smoker. Country of birth was categorized as Scandinavia (Sweden, Denmark, Norway, or Finland), non-Scandinavian Europe, Asia, and other. The MiniMeal-Q food frequency questionnaire[14] was used to estimate total energy intake and energy-adjusted fiber intake. Energy-adjusted fiber intake was determined by calculating the amount of consumed fiber per 1000 kcal of energy. Diet data from women reporting

energy intake values < 500 or >5000 kcal/day, and men reporting <550 or >6000 kcal/day were excluded as they were likely misreported.

Urinary sodium (Architect c16000; Abbott Laboratories, Abbott Park, IL, USA) and creatinine (enzymatic method) were analyzed by Clinical Chemistry at Uppsala University Hospital, Uppsala, Sweden, from the morning fasting spot urine samples collected at the first visit[15,16]. As a surrogate for sodium intake, the 24-h sodium excretion was estimated based on the spot urinary sodium using the Kawasaki formula[17]. Because of the skewed distribution, we used a natural logarithm transformation on sodium intake for adjustment in the regression model.

Information on antibiotics use was extracted from the Swedish Prescribed Drug Register 6 months preceding the visit 1. Proton pump inhibitor (PPI) use was defined as participants with measurable omeprazole and/or pantoprazole levels in plasma from metabolomics data.

## Statistics and reproducibility

**Main analysis**. All the statistical methods were performed using R (version 4.3.1). We used the distance-based multivariate analysis of variance (D-MANOVA) on the species Bray-Curtis dissimilarity matrix to quantify the proportion of variance explained in gut microbiome composition from blood pressure measures (GUniFrac R package). A baseline model was first constructed including age, sex, country of birth, technical source of variation stemming from DNA extraction plates where the samples from the two study centers (Uppsala and Malmö) were analyzed on different plates, smoking, fiber intake, total energy intake, estimated sodium intake, use of antidiabetic medication, use of antihyperlipidemic medication, and body mass index (BMI). Office SBP and DBP were then added, in separate models, to assess their contribution to microbiome variance, followed by the inclusion of the respective 24-h BP. The proportion of variance explained (pseudo-$R^2$) was estimated for each model, and statistical significance was assessed for each model comparison using a pseudo-F-test. Four different 24-h BP outcomes were assessed: mean SBP and DBP, and variability of SBP and DBP, which is measured as the SD of the respective trait. A series of linear regression models were applied in the ABPM subsample to assess the association of microbiome diversity (Shannon diversity index) and species with these four outcomes, one at a time. A false discovery rate (FDR) of 5% using Benjamini–Hochberg method was applied based on P values from each phenotype[18].

We first applied a model with adjustment for age, sex, country of birth, smoking, fiber intake, total energy intake, estimated sodium intake, use of antidiabetic medication, use of antihyperlipidemic medication, BMI, and technical source of variation in Model 1. The selection of these covariates was made using d-separation criteria applied on a directed acyclic graph assisted by the DAGitty, version 3.0, software (www.dagitty.net; Supplementary Fig. S2)[19]. In Model 2, all covariates from Model 1 were included except BMI. We assessed the enrichment for genera and GMMs using the gene set enrichment analysis method[20] from the fgsea R package. For the enrichment analysis, the ranked P values of the associations between species and 24-h BP outcomes were stratified by effect direction from Model 1.

We used a complete case approach, analyzing the 3695 individuals with complete data on these phenotypes. To validate the findings and assess the generalizability of associated species, we repeated the analysis in Model 1 using office BP measurements in the non-ABPM subsample. Partial Spearman correlations were used for the association between 1302 metabolites and BP-signature species in Model 1 with adjustment for age, sex, country of birth, technical source of variation, metabolomics delivery batch, and BMI in those 5742 individuals who had metabolomics analysis available from the 6465 participants included in the main analysis. The 10 annotated metabolite associations with a BP-signature species with the lowest adjusted P values were selected for visualization.

**Sensitivity analysis**. To ensure the associations were not driven by influential observations, unscaled dfbeta values were calculated using dfbeta's R function for each association. For an association to be

considered reliable, the P value after excluding the most influential value had to be <0.05, and the direction of the regression coefficient had to remain unchanged. The following sensitivity analyses were performed in Model 1: (1) exclusion of participants with a dispensed antibiotics prescription within 6 months before the visit 1; (2) exclusion of participants with inflammatory bowel disease; (3) adjustment for PPI usage; (4) additional adjustment for 10 first principal components derived from genotyping.

## Reporting summary

Further information on research design is available in the Nature Portfolio Reporting Summary linked to this article.

## Results

### Baseline characteristics

The ABPM subsample included 2937 participants recruited in Uppsala (mean age 57.3 (SD 4.4) years, 24-h SBP/DBP 122 (10.8)/76 (7.4) mmHg) and 758 participants recruited in Malmö (57.0 (4.3) years, 24-h SBP/DBP 122 (11.8)/75 (8.0) mmHg) (Table 1). The non-ABPM subsample consisted of 195 participants from Uppsala (57.5 (4.7) years, office SBP/DBP, 124 (15.9) /77 (10.7) mmHg) and 2575 participants from Malmö (56.9 (4.2) years, office SBP/DBP, 120 (15.9)/73 (9.5) mmHg) (Table 1). Self-reported comorbidities were similar in Uppsala and Malmö, but a higher proportion of current smokers was noted in Malmö (14.4% in ABPM subsample and 17.9% in non-ABPM subsample) than in Uppsala (8.2% in ABPM subsample and 7.7% in non-ABPM subsample).

### Contribution of office and 24-h blood pressure to gut microbiome variance

D-MANOVA analysis showed that office BP variables explained a small proportion of gut microbiome variance in addition to the baseline model (baseline pseudo-$R^2$: 5.58%; baseline and office BP: 5.62% for SBP, $p_{increment} = 0.003$ and 5.63% for DBP, $p_{increment} < 0.001$). Adding 24-h SBP and 24-h DBP to these models further increased variance explained (5.67% for 24-h SBP, $p_{increment} < 0.001$, 5.66% for 24-h DBP, $p_{increment} < 0.020$).

### Gut microbiota alpha diversity is negatively associated with diastolic blood pressure variation

The association between alpha diversity (Shannon diversity index) and blood pressure outcomes was examined in both ABPM and non-ABPM subsamples using two adjusted statistical models. In Model 1, increased alpha diversity was associated with decreased 24-h DBP variability ($\beta = -0.32$ 95% CI: $-0.59$ to $-0.06$, Table 2). No association was detected with 24-h SBP and DBP, SBP variability, or office BP in Model 1. However, in Model 2, which did not include BMI as a covariate, increased alpha diversity was associated with lower 24-h SBP and DBP, reduced SBP and DBP variability, and lower office SBP and DBP in both the ABPM and non-ABPM subsamples.

### Specific metagenomics species were associated with blood pressure

We further explored the associations between 268 microbial species and 24-h BP outcomes (Fig. 1). In Model 1, which included an adjustment for BMI, we identified *Dysosmobacter* sp001916835 as negatively associated with 24-h SBP and 24-h DBP, *Streptococcus* sp001556435 positively associated with 24-h SBP, and *Intestinimonas massiliensis* negatively associated with 24-h SBP (Supplementary Data 1). No genera or GMMs were enriched for associations after accounting for multiple testing at 5% FDR (Supplementary Data 2 and 3).

In Model 2, which excluded BMI as a covariate, 76 species were associated with 24-h SBP, and 51 species were associated with 24-h DBP. The top finding for negative associations with both SBP and DBP was *Intestinimonas massiliensis*. Conversely, among the top species positively associated were *Streptococcus* sp001556435 for SBP and *Faecalibacterium prausnitzii* for DBP (Supplementary Data 4). Office BP data from the

non-ABPM subsample replicated the association of *Intestinimonas massiliensis* with SBP and *Dysosmobacter* sp001916835 with DBP (Fig. 2). The comparison of estimates in Model 1(BMI-adjusted) and Model 2 (BMI-unadjusted) is shown in Supplementary Fig. S3.

## Specific metagenomics species were associated with blood pressure variability

In Model 1, which included an adjustment for BMI, ER4 sp900317525, from the family *Oscillospiraceae*, was negatively associated with SBP variability, while no species were found to be associated with DBP variability (Supplementary Data 1). In Model 2, 52 species were associated with SBP variability, with ER4 sp900317525 as the top finding for negative association and *Blautia A wexlerae* for positive associations (Supplementary Data 4). Additionally, 18 species were associated with DBP variability in Model 2, with ER4 sp900317525 as the top finding for negative association and *Roseburia intestinalis* for positive association (Supplementary Data 4). No genera or GMMs were enriched for associations after accounting for multiple testing at 5% FDR (Supplementary Data 2 and 3).

## Sensitivity analyses confirmed robust associations between gut microbial species and blood pressure

We performed sensitivity analyses to assess the robustness of the associations between species and 24-h BP outcomes. For associated species, removing the most influential observation provided similar coefficients and $P$ values < 0.05. Analyses excluding participants who had used antibiotics in the previous 6 months or had inflammatory bowel disease, adjustment for PPI use, or genetic principal components provided consistent estimates and $P$ values (Supplementary Data 5 and Supplementary Fig. S4).

## Species associated with 24-h BP were linked to a similar pattern of metabolites

We assessed the associations between the 4 BP-associated species from Model 1 with 1302 plasma metabolites using the partial Spearman correlation, adjusting for age, sex, BMI, and technical source of variation (Supplementary Data 6). The 10 most strongly associated annotated metabolites are presented in Supplementary Fig. S5, showing a similar metabolite association pattern for the 3 species associated with lowered BP and BP

**Table 1 | Descriptive characteristics of participants in Swedish CArdioPulmonary bioImage Study (SCAPIS) in the Uppsala and Malmö centers**

| | ABPM subsample | | non-ABPM subsample | |
| --- | --- | --- | --- | --- |
| | Uppsala (*n* = 2937) | Malmö (*n* = 758) | Uppsala (*n* = 195) | Malmö (*n* = 2575) |
| Age, years | 57.3 (4.4) | 57.0 (4.3) | 57.5 (4.7) | 56.9 (4.2) |
| Female, *n* (%) | 1556 (53.0%) | 403 (53.2%) | 103 (52.8%) | 1418 (55.1%) |
| Country of birth, *n* (%) | | | | |
| Scandinavia | 2628 (89.5%) | 568 (74.9%) | 170 (87.2%) | 2064 (80.2%) |
| Europe | 124 (4.2%) | 128 (16.9%) | 12 (6.2%) | 324 (12.6%) |
| Asia | 118 (4.0%) | 45 (5.9%) | 10 (5.1%) | 129 (5.0%) |
| Other | 67 (2.3%) | 17 (2.2%) | 3 (1.5%) | 58 (2.3%) |
| Current smoker, *n* (%) | 241 (8.2%) | 109 (14.4%) | 15 (7.7%) | 462 (17.9%) |
| Body mass index, kg/m$^2$ | 26.4 (4.1) | 26.7 (4.2) | 27.0 (4.3) | 26.7 (4.3) |
| Fiber intake, g/1000 kcal[a] | 18.6 [13.3;25.5] | 17.9 [12.4;25.8] | 19.6 [13.6;25.4] | 17.9 [12.3;24.9] |
| Total energy intake, kcal/day[a] | 1633 [1298;2050] | 1612 [1236;2112] | 1640 [1227;2052] | 1594 [1244;2078] |
| Diabetes, *n* (%) | 58 (2.0%) | 11 (1.5%) | 6 (3.1%) | 57 (2.2%) |
| Medication for diabetes, *n* (%) | 49 (1.7%) | 10 (1.3%) | 3 (1.5%) | 44 (1.7%) |
| Antihyperlipidemic medication, *n* (%) | 85 (2.9%) | 21 (2.8%) | 6 (3.1%) | 88 (3.4%) |
| Proton pump inhibitor, *n* (%) | 54 (1.8%) | 23 (3.3%) | 3 (1.5%) | 68 (3.5%) |
| Antibiotics treatment[b], *n* (%) | 282 (9.6%) | 71 (9.4%) | 20 (10.3%) | 284 (11.0%) |
| Inflammatory bowel disease, *n* (%) | 39 (1.3%) | 1 (0.1%) | 3 (1.5%) | 30 (1.2%) |
| 24-h BP record | | | | |
| SBP (mmHg) | 122 (10.8) | 122 (11.8) | — | — |
| DBP (mmHg) | 76 (7.4) | 75 (8.0) | — | — |
| Variability of SBP (mmHg) | 15 (4.0) | 14 (3.7) | — | — |
| Variability of DBP (mmHg) | 12 (3.4) | 11 (3.0) | — | — |
| Office BP measurement | | | | |
| SBP (mmHg) | 123 (15.1) | 121 (16.5) | 124 (15.9) | 120 (15.9) |
| DBP (mmHg) | 76 (9.5) | 74 (9.9) | 77 (10.1) | 73 (9.5) |
| Urine sodium (mmol/L)[a] | 55.3 [33.3;86.2] | 47.5 [24.7;76.9] | 52.5 [33.1;86.0] | 44.7 [24.1;76.3] |
| Creatinine (mmol/L)[a] | 124 [73.5;180] | 113 [54.6;163] | 137 [78.1;187] | 109 [54.3;164] |
| Estimated 24-h urine sodium (mg/day)[ac] | 3119 [2359;3970] | 3147 [2395;4049] | 3048 [2297;3967] | 3081 [2375;3919] |

Participants had metagenomics data and were not taking blood pressure medication. Blood pressure measurements were conducted in two subsamples, the ambulatory blood pressure monitoring (ABPM) and non-ABPM subsamples.
*ABPM* ambulatory blood pressure monitoring, *BP* blood pressure, *SBP* systolic blood pressure, *DBP* diastolic blood pressure, *n* number of participants. Continuous variables are presented as mean (standard deviation) or median (interquartile range) as appropriate.
[a]The median and quartile 1 and 3 are presented.
[b]Antibiotics treatment in the previous 6 months.
[c]Estimated 24-h urine sodium was based on Kawasaki formula.

**Table 2 | Association between alpha diversity (Shannon diversity index) and blood pressure (24-h BP outcomes and office BP) in the ABPM and non-ABPM subsamples**

| Outcome | Model 1 | | Model 2 | |
|---|---|---|---|---|
| | Size | Estimate (95% CI) | Size | Estimate (95% CI) |
| ABPM subsample | | | | |
| 24-h SBP | 3695 | −0.41 (−1.24, 0.42) | 3695 | −1.60 (−2.48, −0.73) |
| 24-h DBP | 3695 | −0.29 (−0.87, 0.28) | 3695 | −0.94 (−1.53, −0.34) |
| Variability of 24-h SBP | 3695 | −0.30 (−0.62, 0.02) | 3695 | −0.47 (−0.79, −0.16) |
| Variability of 24-h DBP | 3695 | −0.32 (−0.59, −0.06) | 3695 | −0.49 (−0.76, −0.22) |
| Non-ABPM subsample | | | | |
| Office SBP | 2770 | −1.27 (−2.69, 0.15) | 2770 | −2.70 (−4.15, −1.26) |
| Office DBP | 2769 | −0.44 (−1.31, 0.42) | 2769 | −1.59 (−2.49, −0.70) |

*BP* blood pressure, *ABPM* ambulatory blood pressure monitoring, *SBP* systolic blood pressure, *DBP* diastolic blood pressure, *CI* confidence interval.
Model 1: Adjusted for age, sex, country of birth, and technical source of variation, smoking, fiber intake, total energy intake, sodium intake, usage of antidiabetic drugs, usage of antihyperlipidemic drugs, and body mass index. Model 2: Adjusted for age, sex, country of birth, and technical source of variation, smoking, fiber intake, total energy intake, sodium intake, usage of antidiabetic drugs, and usage of antihyperlipidemic drugs.

variability, respectively: *I. massiliensis*, ER4 sp900317525, and *Dysosmobacter* sp001916835, e.g., with negative associations with microbial metabolites isoursodeoxycholate, imidazole propionate, and positive associations with microbial metabolites p-cresol sulfate and phenylacetylglutamine. The metabolite association pattern was inverse for *Streptococcus* sp001556435, a species positively associated with SBP.

## Discussion

The current study of 3695 participants is the first large population-based study to describe the relationship of the human gut microbiota with 24-h ABPM. Analysis revealed inverse correlations between microbial alpha diversity and DBP variability over 24 h after adjustment for covariates, including BMI. At the species level, we identified four bacterial species associated with ABPM measurements. *Streptococcus* sp001556435 was associated with higher SBP, *Intestinimonas massiliensis* with lower SBP, *Dysosmobacter* sp001916835 with lower SBP and DBP, and ER4 sp900317525 with lower SBP variability. Metabolomic analysis of these four BP-signature species revealed correlations with specific microbial-derived plasma metabolites, particularly isoursodeoxycholate, p-cresol sulfate, and imidazole propionate, suggesting potential mechanistic pathways linking gut microbiota to blood pressure regulation.

The current GTDB representative species for *Streptococcus* sp001556435 corresponds to *S. salivarius* in the NCBI nomenclature. Higher abundance of *S. salivarius* in the oral microbiota was associated with an increased risk of incident hypertension in 1215 postmenopausal women[21]. Similarly, *Streptococcus* spp. was associated with higher SBP in the HELIUS study, which used 16S rRNA gene amplicon sequencing to examine the gut microbiota of 4672 participants from the municipality of Amsterdam[8]. Similarly, *Streptococcus* sp001556435 was associated with higher BP in the present study. In the HELIUS study, increased *Roseburia* spp. and *R. hominis* abundance were inversely associated with SBP. However, in the present study, we found a positive association between *R. inulinivorans* and SBP in the statistical model, not adjusting for BMI (Model 2). This divergence in findings between the HELIUS study and ours could be due to differences in the ethnic groups under investigation and differences between individual species within the genus *Roseburia*.

*Intestinimonas massiliensis*, which was associated with lower SBP in the present study, was inversely associated with prevalent left ventricular diastolic dysfunction in 1996 participants from the Hispanic Community Health Study/Study of Latinos[22]. Hypertension is a well-established cause of left ventricular diastolic dysfunction[23]. The production of SCFA by the gut microbiota has been suggested to affect the host BP[24–27]. *I. massiliensis* can produce the SCFA butyrate[28]. Similarly, *Dysosmobacter* sp001916835, which was associated with lower SBP and DBP in the present study, belongs to *Oscillospiraceae* family, also a known butyrate producer. However,

increased fecal concentrations of SCFA have been described in subjects with elevated SBP in the HELIUS study[8] as well as in other studies[26,27]. Despite the positive associations between serum and fecal SCFA and BP in some observational studies, animal studies have produced opposite results. Oral supplementation of acetate led to a reduction in SBP and DBP in hypertensive mice models[29], and intracolonic or intravenous administration of butyrate produced a reduction in BP in rats[30].

We found that an increased gut microbiome alpha diversity was associated with lower DBP variability. Additionally, we found a member of the *Oscillospiraceae* family, ER4 sp900317525, to be negatively associated with SBP variability after adjusting for BMI, while no species was found to be associated with DBP variability after adjusting for BMI. *Oscillospiraceae* are known butyrate producers and have previously been negatively associated with obesity and inflammatory bowel disease[31,32]. Our findings add to these results, proposing an inverse BMI-independent association with blood pressure variability. Since BP variability is recognized as an independent risk factor for cardiovascular events[33], this finding indicates the potential importance of the gut microbiome in cardiovascular health. A previous study involving 69 participants revealed that higher levels of *Alistipes finegoldii* and *Lactobacillus* spp. were associated with lower BP variability, while *Prevotella* and *Clostridium* spp. correlated with higher variability[34].

The study revealed striking differences in the effect estimates between Model 1 (BMI-adjusted) and Model 2 (BMI-unadjusted), where the BMI adjustment attenuated all assessed associations. The attenuation was more prominent for 24-h SBP and 24-h DBP phenotypes than for the variability of SBP and DBP phenotypes. The substantial differences observed between BMI-adjusted and unadjusted results can be attributed to two key factors. First, BMI may serve as a proxy indicator for lifestyle factors or unmeasured confounding variables that simultaneously influence both microbial species composition and blood pressure[35]. Second, since certain microbial species can potentially affect host adiposity[36], BMI might function as a mediator in the relationship between gut microbiota and blood pressure regulation. Given the well-established causal relationship between elevated BMI and hypertension[37–39], future longitudinal studies with comprehensive data collection are essential to elucidate whether specific gut microbiota species influence blood pressure through BMI-mediated pathways.

The present study has several strengths. This is the first large population-based study to investigate the relationship of the gut microbiota with 24-h ABPM measurements. As elaborated above, 24-h ABPM provides a more precise and comprehensive assessment of BP compared to office measurements. The identified species-BP associations were more strongly associated with 24-h BP compared to office BP, and 24-h BP added information to the multivariable model, including office BP. Additionally, the gut microbiota was assessed using deep shotgun metagenomics, enabling species-level resolution detection of

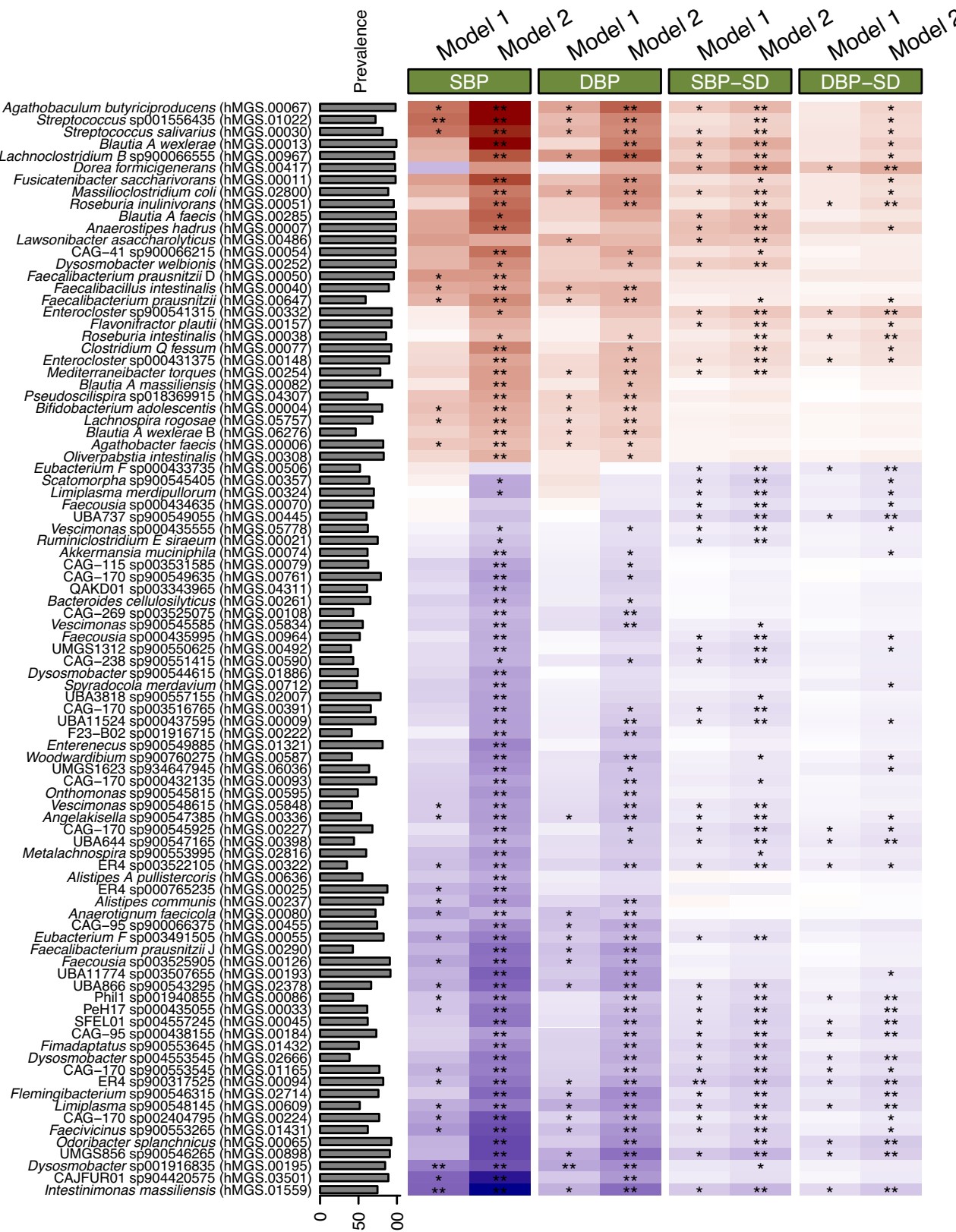

**Fig. 1 | Association between metagenomic species and 24-h blood pressure traits.** Association between metagenomic species and 24-h blood pressure measurements and their variability from multivariable regression analyses in 3695 individuals from the SCAPIS study is shown. In Model 1, adjustment was made for age, sex, country of birth, technical variation, smoking, fiber intake, total energy intake, estimated sodium intake, use of antidiabetic medication, use of antihyperlipidemic medication, and body mass index. Model 2 was adjusted for Model 1 covariates except BMI. Y-axis shows the species name, and the internal identification number (hMGS.number) in parentheses. The prevalence of the metagenomic species in all samples is presented. The color scale ranges from −1 (dark blue, indicating higher negative correlation) to 1 (dark red, indicating higher positive correlation), with white representing no correlation (0). Abbreviations: SBP 24-h systolic blood pressure, DBP 24-h diastolic blood pressure, SBP-SD variability of 24-h SBP, DBP-SD variability of 24-h DBP. ** indicates associations with FDR < 5%, while * indicates associations with P value < 0.05.

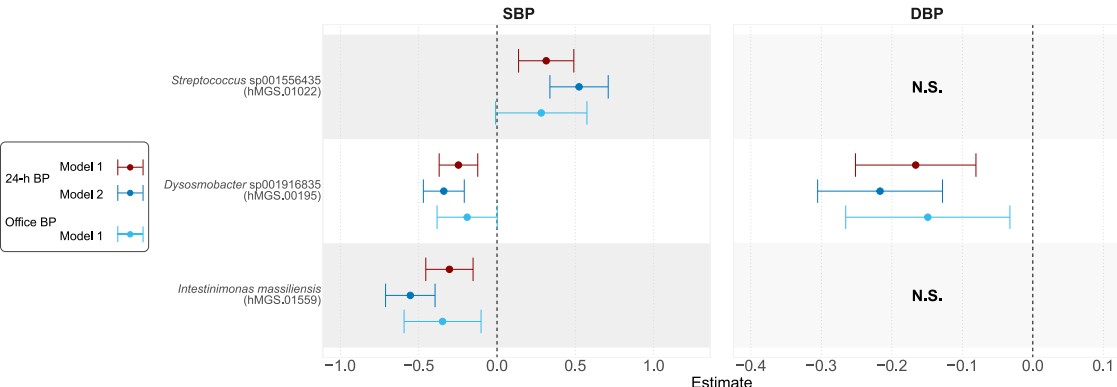

**Fig. 2 | Forest plot of the association between species and blood pressure in two subsamples.** Associations between species and the ambulatory blood pressure monitoring (ABPM) (24-h blood pressure (BP), $n = 3695$) and non-ABPM (office BP, $n = 2770$) subsamples are shown. Model 1 was adjusted for age, sex, country of birth, technical variation, smoking, fiber intake, total energy intake, sodium intake, usage of antidiabetic medications, usage of antihyperlipidemic medications, and body mass index. Each point represents the model coefficient along with the 95% confidence interval. Abbreviations: SBP systolic blood pressure, DBP diastolic blood pressure. N.S. referred to the non-significant associations between species and BP in Model 1.

gut microbiota composition. Moreover, the extensive survey of covariates made it possible to adjust for several potential confounders. We also excluded participants with antihypertensive medication usage, defined by ATC codes, to avoid identifying associations confounded by hypertension treatment.

There are, however, limitations that need to be considered. First, this study has a cross-sectional design, and causality cannot be inferred. Moreover, we cannot rule out collider bias by selection on health-seeking behavior, residual confounding, or reverse causation. For instance, participants with higher blood pressure might have received recommendations of lifestyle modifications that could affect the gut microbiota. Second, formal mediation analysis for BMI was not possible due to lack of temporal precedence in the present study. We expect to be able to address the longitudinal associations between gut microbiota and BP in follow-up examinations of SCAPIS. Third, the dietary assessment was based on a self-reported questionnaire that may be affected by recall and reporting bias. Misclassification of dietary intake has been related to factors that are relevant for the gut microbiota composition, such as sex, age, or obesity[40]. Therefore, we cannot exclude the possibility of differential misclassification of the dietary covariates dependent on the exposure, which would result in residual confounding or additional distortion of estimates. Fourth, while we replicated the associations of *Intestinimonas massiliensis* with SBP and *Dysosmobacter* sp001916835 with DBP using office BP in an independent subsample, several validation aspects remain important. Our findings regarding BP variability would benefit from replication in cohorts with 24-h BP data, as this measurement cannot be captured through office visits. Future longitudinal studies with repeated microbiota sampling and BP measurements would be particularly valuable for determining whether these microbial associations are stable over time or represent transient fluctuations, and for clarifying potential causal pathways, especially regarding BMI as a possible mediator. Fifth, although we estimated salt intake using urine sodium from spot samples, this method has been shown to have individual-level inconsistencies when compared with 24-h urine samples[41]. Sixth, fecal metagenomics, while powerful for describing a snapshot of the microbial community, does not accurately capture the gut microbiota species that reside closer to the mucosal membrane[42] or in the small intestines[43], which is a limitation of most population-based studies on gut microbiota.

In conclusion, in our large population-based sample, the associations between the gut microbiota, plasma metabolites, and 24-h BP were investigated. We identified four gut microbiota species associated with 24-h ABPM. Our findings on the role of human gut microbiota and its downstream metabolites in BP regulation provide potential targets for future research.

## Data availability

SCAPIS data are not publicly available due to privacy and ethical restrictions. Access to pseudonymized SCAPIS phenotype and genotype data can be applied for from the SCAPIS Data Access Board (https://www.scapis.org/data-access/) and requires ethical approval from the Swedish Ethical Review Board. The source data underlying all figures used to generate the results are available at https://github.com/MolEpicUU/24hBP-mgs.

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

## Acknowledgements

The computations and data handling were enabled by resources in project sens2019512 provided by the National Academic Infrastructure for Supercomputing in Sweden (NAISS) at UPPMAX, funded by the Swedish Research Council through grant agreement no. 2022-06725. The main funding body of The Swedish CArdioPulmonary bioImage Study (SCAPIS) is the Swedish Heart and Lung Foundation. The study is also funded by the Knut and Alice Wallenberg Foundation, the Swedish Research Council, VINNOVA (Sweden's Innovation agency), the University of Gothenburg and Sahlgrenska University Hospital, Karolinska Institutet and Region Stockholm, Linköping University and University Hospital, Lund University and Skåne University Hospital, Umeå University and University Hospital, Uppsala University and University Hospital. We would like to acknowledge the help of Biobank Sweden and the local biobank facilities for their services in handling of biological samples and biobanking. We acknowledge the financial support from the European Research Council [ERC-STG-2018-801965 (T.F.); ERC-CoG-2014-649021 (M.O-M.), ERC-STG-2015-679242 (J.G.S.)], the Swedish Research Council [VR 2019-01471 (T.F.); 2018-02784 (M.O-M.); 2018-02837 (M.O-M.); 2021-03291 (M.O-M.); EXODIAB 2009-1039 (M.O-M.); 2019-01015 (J.Ä.); 2020-00243 (J.Ä.); 2019-01236 (G.E.); 2021-02273 (J.G.S.)], the Swedish Heart-Lung Foundation [Hjärt-Lungfonden, 2023-0687 (T.F.); 20200711 (M.O-M.); 20180343, 20210357 (J.Ä.); 20200173 (G.E.); 20190526 (J.G.S.)], the A.L.F. governmental grant [2018-0148 (M.O-M.)], the Novo Nordic Foundation [NNF20OC0063886 (M.O-M.)], the Swedish Diabetes foundation [DIA 2018-375 (M.O-M.)], the Swedish Foundation for Strategic Research [LUDC-IRC 15-0067 (M.O-M.)].

## Author contributions

J.Ä., J.S., G.Be, G.E., J.G.S., M.O-M., and T.F. obtained the funding. Y-T.L., U.H., S.S-B., J.S., J.Ä., M.O-M., and T.F. designed the study and developed the concept. L.L., J.Ä., G.E., J.G.S., J.S., J.Ä., M.O-M., and T.F. collected the data. N.N., A.C.E., J.B.H., H.B.N. performed metagenomic analysis and bioinformatics, Y-T.L., K.F.D., S.S-B. performed quality control and/or data management, Y-T.L. and S.S-B. performed the association analyses, Y-T.L., K.F.D., S.S-B. and G.V. visualized the results, and M.O-M. and T.F.

coordinated the study. Y-T.L., G.Ba., D.N., and T.F. wrote the first draft of the manuscript. All authors contributed with the interpretation of the results and critical revision of the manuscript.

## Funding

## Competing interests
The authors declare the following competing interests: N.N., A.C.E. and J.B.H., and H.B.N. are employees of Cmbio. The funders had no role in study design, data collection and analysis, decision to publish, nor preparation of the manuscript. J.Ä. has received lecture fees from Novartis and AstraZeneca, and served on advisory boards for AstraZeneca and Boerhinger Ingelheim, all unrelated to the present paper. J.S. reports stock ownership in Anagram kommunikation AB and Symptoms Europe AB, unrelated to the present study. All other authors declare no conflicts of interest in connection with this study.

## Additional information

[1]Molecular Epidemiology, Department of Medical Sciences, Uppsala University, Uppsala, Sweden. [2]Department of Family Medicine, Kaohsiung Medical University Hospital, Kaohsiung Medical University, Kaohsiung, Taiwan, ROC. [3]CIBER Cardiovascular Diseases (CIBERCV), Instituto de Salud Carlos III, Madrid, Spain. [4]Cmbio A/S, Copenhagen, Denmark. [5]Clinical Epidemiology, Department of Medical Sciences, Uppsala University, Uppsala, Sweden. [6]Department of Molecular and Clinical Medicine, Institute of Medicine, Sahlgrenska Academy, University of Gothenburg, Gothenburg, Sweden. [7]Department of Clinical Physiology, Sahlgrenska University Hospital, Gothenburg, Sweden. [8]The Wallenberg Laboratory/Department of Molecular and Clinical Medicine, Institute of Medicine, Gothenburg University and the Department of Cardiology, Sahlgrenska University Hospital, Gothenburg, Sweden. [9]Department of Cardiology, Clinical Sciences, Lund University and Skåne University Hospital, Lund, Sweden. [10]Wallenberg Center for Molecular Medicine and Lund University Diabetes Center, Lund University, Lund, Sweden. [11]Department of Clinical Sciences, Lund University, Malmö, Sweden. [12]Division of Family Medicine and Primary Care, Department of Neurobiology, Care Sciences and Society, Karolinska Institute, Stockholm, Sweden. [13]School of Health and Social Studies, Dalarna University, Falun, Sweden. [14]The George Institute for Global Health, University of New South Wales, Sydney, NSW, Australia. ✉e-mail: tove.fall@medsci.uu.se

