## [Transparent Peer Review file · Communications Medicine]

The association between the gut microbiome and 24-hour blood pressure measurements in the SCAPIS study

Corresponding Author: Professor Tove Fall

Version 0:

Reviewer comments:

Reviewer #1

(Remarks to the Author)

In this study, Lin et al analyze the cross-sectional associations between the gut microbiome and ambulatory blood pressure (BP) traits.

Although this study presents some strengths including the use of ambulatory BP monitoring rather than office BP, the large number of participants (n=4063), and gut metagenomes, I have important major comments:

- It is highlighted that the gut microbiome composition is measured at high resolution, which is not completely right. Although metagenomes have been used, the analyses focus on species. A higher resolution would be to use species-level genome bins (SGBs).
- Based on Supplementary Fig 1, it seems that the microbiome data was prefiltered based on the prevalence. However, this is not specified in the methods. Please add all the prefiltering steps conducted or make the figure clearer. Moreover, it seems that after removing the MGS with <30% non-zero value, 1692 MGS were analyzed, however, supplementary table 1 presents 1692 MGS and some with a very low prevalence.
- BMI is suggested to be a confounder or mediator in the association between BP and the alpha-diversity index. Are you suggesting that the microbiome influences BP by affecting BMI? I find it tricky to have BMI adjustment for the third model rather than including it as a main covariate from model 1.
- When reporting the associations between microbial species and BP traits, it would be nice to mention the microbes that appear significant with most BP traits. "we found 41 species positively associated with 24-hour SBP, 41 with 24-hour DBP, 27 with the variability of SBP, and 17 with the variability of DBP" this does not say that much.
- Line 289: Specify why these 140 species were further analyzed.
- Why is the prevalence missing for some species from Supplementary Table 4?
- The discussion mentions that formal mediation analyses cannot be performed as it is a cross-sectional study. However, mediation analysis can still be conducted while being aware of its limitations.
- The study would benefit from actual metabolomics profiling to be certain of the associations between the identified species and metabolites. Line 528: I would remove plasma metabolites as these actually were not measured and assessed along with the microbiome and BP traits.
- Line 421: "one of the SCFA, specifically butyrate/isobutyrate (4:0), was found to be linked to 59 species associated with BP." While this is stated in the discussion, it is not in the results section. Please discuss what has been stated in the results. Moreover, I am surprised that butyrate/isobutyrate is grouped. Butyrate and isobutyrate are two compounds different and their roles in BP have been reported to be opposite. In Supplementary Table 7, 2-hydroxybutyrate/2-hydroxyisobutyrate appears rather than butyrate/isobutyrate. Please note these are not the same as butyrate/isobutyrate and they have different chemical properties.

Some minor comments to have in mind:

- Line 243: Include SD along with the mean.
- Line 253: Specify that alpha-diversity is what is being investigated.
- Line 274: Add the threshold used.
- Line 289: 'with the exception of Eubacteriales': this statement is unclear.

- Line 334: This paragraph is repeated in line 337.
- Line 344: 'associated with many plasma metabolites' is a very abstract statement.
- Line 374: This is already discussed in the introduction.

Reviewer #2

(Remarks to the Author)

The study elucidates the correlation between the gut microbiome and blood pressure on a large population cohort volunteered from Swedish population. The big advantage of the study is a large sample size and 24-hour DB measurement. But...

1. Among the conclusions, the authors indicated that "species ... showed consistent results across 24-hour BP and office BP". It basically means that there's not a lot of added value in the 24BP compared to office BP - one of the main advantages of this current manuscript claimed by authors in the introduction. Thus, the "selling point" of the paper contradicts with actual results. First, it would make sense to target the manuscript to the traits that are not covered by office BP, such as BP variability. Second, it's worth spending words and time on exploring the divergence between office BP and 24hour BP instead of their similarity. Third, it requires better formulations among the manuscript that don't overclaim 24BP as a super power of the study.
2. Since the cohort is not not monoethnic, the genetic makeup of the participants seems to be under-controlled by considering only the country of birth. If genetic data is available for the participants, it makes sense to adjust for genetic kinship (several genetic PCs or kinship matrix) in the analysis. This could be one of the reasons for possible statistics inflation.
3. It doesn't seem well justified to avoid composition-aware methods for transformation, like CLR, or dirichlet multinomial, or so. This might be the second potential reason for possible statistics inflation.
4. Benjamini-Hochberg procedure for FDR control doesn't perform well in the case of correlated tests which is definitely the case for microbiome data (especially if compositionality is not taken into account, see p.2). The third reason for possible statistics inflation.
5. There's quite a lot of species were associated with the traits of interest. Quite a lot, even too much. It might indicate the inflation towards FPs due to undercorrection for some of important phenotypes, or not-well-fit statistical approach such as too loyal FDR. The problem is partly targeted by authors by post-hoc checkup for outliers, adjusting for extra phenotypes such antibiotics and IBD, but not completely.
6. Results: "We considered BMI as a potential mediator or confounder." Why not trying to address that in more detail via causal inference analysis or so? The associations 'corrupted' by BMI confounder effect are not reliable anyway.
7. I don't get why beta diversity tests are missing. This is an important information, especially in the case of such a strong phenotype that is claimed to be associated with hundreds of species. Otherwise, why no machine learning approaches were applied to see how much of BP variance are explained by antropometrics/BMI or added by microbiome on top of that? This is again much more important information that naming of those hundreds of species and spending discussion on that.
8. Conclusions: "Fourth, our findings would need replication in an independent cohort with complete 24-hour ABPM and gut microbiota data." No, the authors just showed that there's no big difference between 24hAB and office AB, so any replication cohort of appropriate size will be fine (expect for AB variability which is understudied here).
9. Conclusions: "Our findings provide a starting point for further studies". Definitely not: there many, many cohorts which studied the relationship between BP and microbiome. It's too late for many years to claim this work as a 'starting point'.

To conclude:

- More detailed attention should be given to BMI-independent associations and actual added value for 24BP
- Statistics approach/transformations/FDR should be revisited.

Version 1:

Reviewer comments:

Reviewer #1

(Remarks to the Author)

All my concerns have been addressed.

Reviewer #2

(Remarks to the Author)

The authors have done a good job in addressing my comments and comments of my colleague. The key points were the proper adjustment for confounders (BMI/genetics), switching focus on the added value of variable BP, and a bit more advanced composition/FDR control methods. Well done, yet there were a few points that have been missed or misinterpreted that are still worth doing:

"Authors' reply: Thank you for your suggestion. The outcome of this study is blood pressure (BP), so we do not categorize BP into groups. Therefore, we did not perform beta diversity tests for group comparison." - PERMANOVA doesn't require factor or ordinal trait to correlate with dissimilarity matrix. I could

be easily done on a quantitative phenotype as well. And it gives some significant messages, in particular - what's the added value pseudo-R² of (i.e. total microbiome composition variance explained) of variable BP on top of office BP? That can be done by sequential adding of terms in permanova. Adjustment of covariates can also easily be done with that.

"Transformed values originally zero were replaced with the minimal non-zero transformed value for each species." - this would be a reasonable approach for metabolomics. where the detection limit might vary from one entity to another. For the sequencing it's not quite like that, and universal pseudocount might be used instead.

It's also possible to deeper in the discussion on some other points raised by authors in the rebuttal, but I think these two minor aforementioned things could make manuscript great enough for the publication from both technical and scientific perspective

Version 2:

Reviewer comments:

Reviewer #2

(Remarks to the Author)

my minor comments were properly taken into account. I have no more concerns. Congrats with a good paper!

Reviewer #1 (Remarks to the Author):

In this study, Lin et al analyze the cross-sectional associations between the gut microbiome and ambulatory blood pressure (BP) traits.

Although this study presents some strengths including the use of ambulatory BP monitoring rather than office BP, the large number of participants (n=4063), and gut metagenomes, I have important major comments:

1. It is highlighted that the gut microbiome composition is measured at high resolution, which is not completely right. Although metagenomes have been used, the analyses focus on species. A higher resolution would be to use species-level genome bins (SGBs).

Authors' reply: Thank you for the thorough review and insightful comments that helped us improve the manuscript. To increase the species-level profiling quality, the raw sequencing data are now reanalyzed with the most recent Clinical Microbiomics Human Microbiome Profiler (CHAMP) profiler, a modern algorithm described by Pita et al. (2024), in <https://doi.org/10.3389/fmicb.2024.1425489>. We have revised our manuscript to state "species-level resolution" rather than claiming high resolution.

Updated text in the Introduction section:
page 8, lines 11-14

*...**Here**, we aimed to investigate the association of the gut microbiome composition measured at ~~high-resolution~~ **species-level resolution and functional level** with ABPM measurements...*

Updated text in the Discussion section:
page 25-26, lines 19-1

*Additionally, the gut microbiota was assessed using deep shotgun metagenomics, enabling ~~high-resolution~~ **species-level** resolution detection of gut microbiota ~~species~~**composition**...*

2. Based on Supplementary Fig 1, it seems that the microbiome data was prefiltered based on the prevalence. However, this is not specified in the methods. Please add all the prefiltering steps conducted or make the figure clearer. Moreover, it seems that after removing the MGS with <30% non-zero value, 1692 MGS were analyzed, however, supplementary table 1 presents 1692 MGS and some with a very low prevalence.

Authors' reply: We apologize for the lack of clarity and inconsistency in the previous version regarding filtering. We have now re-run all analysis for the 268 species that had a relative abundance >0.01% in >30% of study participants and updated the manuscript and Figure S1 accordingly

Updated text in the method section:
page 11, lines 16-17.

A total of 4594 microbial species were identified. Of these, 268 species met our inclusion criteria of having a relative abundance greater than 0.01% in more than 30% of study participants.

Supplementary Figure S1:

Figure S1. Flowchart of study design

3. BMI is suggested to be a confounder or mediator in the association between BP and the alpha-diversity index. Are you suggesting that the microbiome influences BP by affecting BMI? I find it tricky to have BMI adjustment for the third model rather than including it as a main covariate from model 1.

Authors' reply: We appreciate this valuable point. We cannot exclude that some species are affecting adiposity and in the extension BP traits. However, it is not possible to distinguish BMI from being a mediator from a confounder in this cross-sectional data. In response to the comment, we have revised our analytical strategy to the conservative approach suggested by the Reviewer, where our new primary model (Model 1) now includes BMI adjustment along with other covariates, while Model 2 investigates the associations without BMI adjustment. We have focused on the BMI-adjusted results in the article.

Updated text Method section:
page 14, lines 15-19.

We first applied a model with adjustment for age, sex, country of birth, smoking, fiber intake, total energy intake, estimated sodium intake, use of antidiabetic medication, use of antihyperlipidemic medication, body mass index (BMI) and technical source of variation stemming from DNA extraction plates where the samples from the two study centers (Malmö and Uppsala) were analyzed on different plates in Model 1. The selection of these covariates was made using d-separation criteria applied on a directed acyclic graph assisted by the DAGitty, version 3.0, software (www.dagitty.net; Supplementary Figure S2). In Model 2, all covariates from Model 1 were included except BMI.

4. When reporting the associations between microbial species and BP traits, it would be nice to mention the microbes that appear significant with most BP traits. “we found 41 species positively associated with 24-hour SBP, 41 with 24-hour DBP, 27 with the variability of SBP, and 17 with the variability of DBP” this does not say that much.

Authors' reply: Thank you for your suggestion. In the revised version one species, *Dysosmobacter sp001916835*, was associated with two traits, while the other three were associated with one trait only.

Updated text in the Results section:
page 18, lines 8-12

*In Model 1, which included an adjustment for BMI, we identified *Dysosmobacter sp001916835* as negatively associated with 24-hour SBP and 24-hour DBP, *Streptococcus sp001556435* positively associated with 24-hour SBP, and *Intestinimonas massiliensis* negatively associated with 24-hour SBP (Supplementary Table S1).*

5. Line 289: Specify why these 140 species were further analyzed.

Authors' reply: Thank you for the opportunity to clarify. Based on your suggestions regarding BMI, we adjusted our strategy to assess in both Model 1 and Model 2 all species that passed the abundance/prevalence filtering.

6. Why is the prevalence missing for some species from Supplementary Table 4?

Authors' reply: We apologize for the missing prevalence data. We have ensured that all species have the prevalence reported in Supplementary Table 1 and 2.

7. The discussion mentions that formal mediation analyses cannot be performed as it is a cross-sectional study. However, mediation analysis can still be conducted while being aware of its limitations.

Authors' reply: Thank you for your valuable input. We appreciate your suggestion regarding mediation analysis. While we acknowledge that mediation analysis can technically be conducted in cross-sectional studies, we chose to refrain from performing it in this case as we do not know if BMI is a confounder or mediator. However, we have chosen to present and compare results from Model 1 (with BMI adjustment) and Model 2 (without BMI adjustment) to evaluate the effect of BMI as a covariate. The detailed results are provided in Supplementary Tables 1 and 2, and a comparison in Supplementary Figure S3. From this comparison it appears that the BMI adjustment reduces the beta estimates for all four outcomes, but to a larger degree for SBP and DBP than for SD-traits. We have also expanded our discussion on this topic.

Supplementary Figure S3. Comparison of beta estimates in BMI-adjusted and non-adjusted models for associations with q -value < 0.05 in Model 2.

Updated text in the Discussion section:
page 24-25 lines 16-10.

The study revealed striking differences in the effect estimates between Model 1 (BMI-adjusted) and Model 2 (BMI-unadjusted) where the BMI adjustment attenuated all assessed associations. The attenuation was more prominent for 24h-SBP and 24h-DBP-phenotypes than for the SBP-SD and DBP-SD phenotypes. The substantial differences observed between BMI-adjusted and unadjusted results can be attributed to two key factors: First, BMI may serve as a proxy indicator for lifestyle factors or unmeasured confounding variables that simultaneously influence both microbial species composition and blood pressure.³⁴ Second, since certain microbial species can potentially affect host adiposity,³⁵ BMI might function as a mediator in the relationship between gut microbiota and blood pressure regulation. Given the well-established causal relationship between elevated BMI and hypertension,³⁶⁻³⁸ future longitudinal studies with comprehensive data collection are essential to elucidate whether specific gut microbiota species influence blood pressure through BMI-mediated pathways.

8. The study would benefit from actual metabolomics profiling to be certain of the associations between the identified species and metabolites. Line 528: I would remove plasma metabolites as these actually were not measured and assessed along with the microbiome and BP traits.

Authors' reply: Thank you for this suggestion. We have the metabolomic profiling available in SCAPIS and have now re-analyzed the correlation between the four BP-signature species and 1302 plasma metabolites by partial Spearman correlation adjusting for age, sex, technical variation, country of birth and BMI. The details of the methods regarding sample collection and metabolomic profiling were also added to the method section.

Updated text in the Method section:
page 12, lines 13-19

Plasma metabolites sample collection and processing

Blood specimens were obtained from study participants who fasted overnight prior to their clinical visit. All collected samples underwent plasma separation before being preserved at -80 degrees Celsius within the facility's biorepository before being transported to the analytical laboratory at Metabolon Inc. (Durham, North Carolina)⁵⁸ for comprehensive metabolomic evaluation. Details are given in the Supplementary Methods and in Dekkers et al.¹¹

Page 13. lines 12-16

Partial Spearman correlations were used for the association between 1302 metabolites and BP-signature species in Model 1 adjustment for age, sex, place of birth, metagenomic DNA extraction plate, metabolomics delivery batch and BMI in those 5742 individuals that had metabolomics analysis available from the 6465 participants included in the main analysis. The ten annotated metabolites associations with a BP-signature species with the lowest adjusted P-values were selected for visualization.

Updated text in the Results section:
page 20-21, lines 11-4

Species associated with 24-hour BP were linked to a similar pattern of metabolites
We assessed the associations between the 4 BP-associated species from Model 1 with 1302 plasma metabolites using the partial Spearman correlation adjusting for age, sex, BMI and technical source of variation (Supplementary Table S4). The ten most strongly associated annotated metabolites are presented in Supplementary Figure S5, showing a similar metabolite association pattern for the three species associated with lowered BP and BP variability, respectively: *I. massiliensis*, ER4 sp900317525 and *Dysosmobacter* sp001916835 e.g. with negative associations with microbial metabolites isoursodeoxycholate, imidazole propionate and positive associations with microbial metabolites p-cresol sulfate and phenylacetylglutamine. The metabolite association pattern was inverse for *S. sp001556435*, a species positively associated with SBP.

Supplementary Figure S5. Heatmap showing associations between 24-hour mean BP-associated species and the ten annotated plasma metabolites with lowest p-values. The color scale ranges from -1 (dark blue, indicating strong negative correlation) to 1 (dark red, indicating strong positive correlation), with white representing no correlation (0). Metabolites are categorized into distinct subclasses.

9. Line 421: “one of the SCFA, specifically butyrate/isobutyrate (4:0), was found to be linked to 59 species associated with BP.” While this is stated in the discussion, it is not in the results section. Please discuss what has been stated in the results. Moreover, I am surprised that butyrate/isobutyrate is grouped. Butyrate and isobutyrate are two compounds different and their roles in BP have been reported to be opposite. In Supplementary Table 7, 2-hydroxybutyrate/2-hydroxyisobutyrate appears rather than butyrate/isobutyrate. Please note these are not the same as butyrate/isobutyrate and they have different chemical properties.

Authors’ reply: Thank you for noting this error. We have removed this paragraph because the results of the species and metabolites were changed.

Some minor comments to have in mind:

- Line 243: Include SD along with the mean.

Authors’ reply: SD has been added.

Updated text in the Results section:
page 17, lines 3-11

The ABPM subsample included 2937 participants recruited in Uppsala (mean age 57.3 (4.4) years, mean 24-hour SBP/DBP 122(10.8)/76(7.4) mmHg) and 758 participants recruited in Malmö (mean age 57.0 (4.3) years, mean 24-hour SBP/DBP 122(11.8)/75(8.0) mmHg) (Table 1). The non-ABPM subsample consisted of 195 participants from Uppsala (mean age 57.5(4.7) years, office SBP/DBP, 124(15.9)/77(10.7) mmHg) and 2575 participants from Malmö (mean age 56.9(4.2) years, office SBP/DBP, 120(15.9)/73(9.5) mmHg)...

- Line 253: Specify that alpha-diversity is what is being investigated.

Authors’ reply: Alpha-diversity has been added.

Updated text in the Results section:
page 17, lines 14-15.

Gut microbiota alpha diversity is negatively associated with diastolic blood pressure variation

- Line 274: Add the threshold used.

Authors’ reply:

Updated text in the Results section:
page 18, lines 13-14
Page 19, lines 15-16

No genera or GMMs were enriched for associations after accounting for multiple testing at 5% FDR.

- Line 289: ‘with the exception of Eubacteriales’: this statement is unclear.

Authors’ reply: Thank you for your suggestion. The sentence has been removed because the species has changed.

- Line 334: This paragraph is repeated in line 337.

Authors' reply: The redundant paragraph has been removed.

- Line 344: 'associated with many plasma metabolites' is a very abstract statement.

Authors' reply: We have revised the subtitle. We have also visualized the top 10 metabolites in the heatmap.

Updated text in the Results section:
page 20, lines 11-12.

Species associated with 24-hour BP were linked to a similar pattern of metabolites

- Line 374: This is already discussed in the introduction.

Authors' reply: We have removed this paragraph.

~~*The gold standard for determining the mean BP in an individual is with 24-hour ABPM, as it can account for, and quantify the substantial within-person variation in blood pressure over time. Its clinical practical application is widely acknowledged, and its usefulness for population research has recently been also recognized.¹⁰ Prior to this study, the association between gut microbiota and within-person variation in BP had not been investigated.*~~

Reviewer #2 (Remarks to the Author):

The study elucidates the correlation between the gut microbiome and blood pressure on a large population cohort volunteered from Swedish population. The big advantage of the study is a large sample size and 24-hour DB measurement. But...

1. Among the conclusions, the authors indicated that “species ... showed consistent results across 24-hour BP and office BP”. It basically means that there’s not a lot of added value in the 24BP compared to office BP - one of the main advantages of this current manuscript claimed by authors in the introduction. Thus, the “selling point” of the paper contradicts with actual results. First, it would make sense to target the manuscript to the traits that are not covered by office BP, such as BP variability. Second, it’s worth spending words and time on exploring the divergence between office BP and 24hour BP instead of their similarity. Third, it requires better formulations among the manuscript that don’t overclaim 24BP as a super power of the study.

Authors’ reply: Thank you for these insightful comments. This shift in focus would better highlight the added value of 24-hour BP monitoring. We have restructured the results and discussion sections to emphasize findings related to BP variability, as this aspect differentiates the study from those using only office BP measurements.

Updated text in the Results section:
page 19, lines 6-16

Specific metagenomics species was associated with blood pressure variability
In Model 1, which included an adjustment for BMI, ER4 sp900317525, from the family Oscillospiraceae, was negatively associated with SBP variability, while no species were found to be associated with DBP variability (Supplementary Table S1). In Model 2, 52 species were associated with SBP variability, with ER4 sp900317525 as the top finding for negative association and Blautia A wexlerae for positive associations (Supplementary Table S2). Additionally, 18 species were associated with DBP variability in Model 2, with ER4 sp900317525 as the top finding for negative association and Roseburia intestinalis for positive association (Supplementary Table S2). No genera or GMMs were enriched for associations after accounting for multiple testing at FDR <5% (Supplementary Table S5, S6).

Updated text in the Discussion section:
page 24, lines 4-16

We found that an increased gut microbiome alpha diversity was associated with lower DBP variability. Additionally, we found a member of the Oscillospiraceae family, ER4 sp900317525, to be negatively associated with SBP variability, while no species was found to be associated with DBP variability after adjusting for BMI. Oscillospiraceae are known butyrate producers and has previously been negatively associated with obesity and inflammatory bowel disease^{32 33}. Our findings add to these results proposing an inverse BMI-independent association with blood pressure variability. Since BP variability is recognized as an independent risk factor for cardiovascular events,³⁰ this finding indicates the potential importance of the gut microbiome in cardiovascular health. A previous study involving 69 participants revealed that higher levels of Alistipes finegoldii and Lactobacillus spp. were associated with lower BP variability, while Prevotella and Clostridium spp. correlated with higher variability³¹.

2. Since the cohort is not monoethnic, the genetic makeup of the participants seems to be under-controlled by considering only the country of birth. If genetic data is available for the participants, it makes sense to adjust for genetic kinship (several genetic PCs or kinship matrix) in the analysis. This could be one of the reasons for possible statistics inflation.

Authors' reply: We have now conducted a sensitivity analysis including genetic principal components based on genotyping into our models to assess the potential impact of population structure on our results. We found similar results in this analysis and have added the results in Supplementary Table S4 and Supplementary Figure S3.

Updated text Methods section
page 16, lines 5-9

The following sensitivity analyses were performed in Model 1: (1) exclusion of participants with a dispensed antibiotic prescription within 6 months before the visit 1; (2) exclusion of participants with inflammatory bowel diseases; (3) adjustment for PPI usage; and (4) additional adjustment for 10 first principal components derived from genotyping.

page 20, lines 3-9

We performed sensitivity analyses to assess the robustness of the associations between species and 24-hour BP outcomes. For associated species removing the most influential observation provided similar coefficients and P -values < 0.05 . Analyses excluding participants who had used antibiotics in the previous 6 months or had inflammatory bowel diseases, adjustment for PPI use or genetic principal components provided consistent estimates and P -values (Supplementary Table S3, Supplementary Figure S4).

Figure S4. Sensitivity analyses. Association between results from the Model 2 with 24-hour blood pressure outcomes in which participants who underwent antibiotic treatment (Antib) during the previous 6 months before visit 1, had inflammatory bowel diseases (IBD), had proton-pump inhibitors (PPI) measured in plasma, were excluded. Additional adjustments were made for genetic principal components to account for population stratification by adding the first 10 principal components (PCs:1-10).

3. It doesn't seem well justified to avoid composition-aware methods for transformation, like CLR, or dirichlet multinomial, or so. This might be the second potential reason for possible statistics inflation.

Authors' reply: We agree that our approach did not fully account for the compositional nature of microbiome data. We have now applied a CLR transformation and rerun the analysis.

Updated text Method section:
page 11-12 lines 19-3

Species abundances underwent centered-log ratio (clr) transformation, with a pseudo-value added to address zero-abundance challenges. Transformed values originally zero were replaced with the minimal non-zero transformed value for each species.

4. Benjamini-Hochberg procedure for FDR control doesn't perform well in the case of correlated tests which is definitely the case for microbiome data (especially if compositionality is not taken into account, see p.2). The third reason for possible statistics inflation.

Authors' reply: Thank you for the suggestion. We have now taken compositionality into account by changing the method to CLR. We agree with the Reviewer that correlated tests could be an issue for the Benjamini-Hochberg procedure. However, in our opinion the literature suggests that potential bias will be in a conservative direction. Thus, we would still ensure control of the false discovery rate. We quote from Benjamini (2010) (doi/epdf/10.1111/j.1467-9868.2010.00746.x): "Independence of test statistics was assumed in Benjamini and Hochberg (1995). Addressing positive dependence by Benjamini and Yekutieli (2001) was essential in assuring users that the simple procedure in Benjamini and Hochberg (1995) was safe to use in many situations arising in practice. It built on the work of Sarkar (1998) and was followed by the work of Sarkar, Finner and co-workers. The modification to general dependence is often not needed: convincing simutheoretical evidence indicates that the same holds for two-sided z-tests with any correlation structure (Reiner-Benaim, 2007), but the theory awaits a complete proof." A partial, but not full, proof of Benjamini-Hochberg controlling the false discovery rate for two-sided tests for arbitrary correlation structures has been recently presented by Sarkar (2023) (<https://arxiv.org/abs/2304.05261>).

To ensure that our use of the Benjamini-Hochberg procedure is not a concern, we conducted simulations according to the following procedure: First, we reshuffled the values of systolic blood pressure, thus making the variable independent of all our species, while also preserving any remaining species correlation. We also generated 50 multivariate normal random variables in blocks of 5, with a Pearson correlation of 0.5 within each block. All these variables were associated with the reshuffled systolic blood pressure. Then we regressed systolic blood pressure on all of our species variable and the 50 new variables, one at the time. Adjustment variables were identical to the variables used for model 1 (with full adjustment) in the paper. We then stored the p-values from all regressions and performed a Benjamini-Hochberg multiple correction on them with a cut-off of 5%. We iterated this procedure 1000 times. The average false discovery rate in our simulations was 4.0%, which indicated that, at least for our simulation setting, the FDR was slightly conservative. We also repeated the simulation settings without the 50 multivariate normal variables (i.e. the null hypothesis was true for every analysis). In these simulations, there were no false positives in any of the 1000 iterations.

5. There's quite a lot of species were associated with the traits of interest. Quite a lot, even too much. It might indicate the inflation towards FPs due to undercorrection for some of important phenotypes, or not-well-fit statistical approach such as too loyal FDR. The problem is partly targeted by authors by post-hoc checkup for outliers, adjusting for extra phenotypes such antibiotics and IBD, but not completely.

Authors' reply: Thank you for raising this important concern regarding the potential inflation of false positives due to undercorrection or suboptimal statistical approaches. In response to feedback from both Reviewers, we have taken several steps to re-analyze the data:

- We revised our data transformation approach to apply the centered log-ratio (CLR) transformation, which better accounts for the compositional nature of microbiome data.
- We adjusted our analysis strategy to include two models: Model 1 (fully adjusted, including BMI) and Model 2 (without BMI adjustment), both using the full set of metagenomic species.
- We reanalyzed the data with higher resolution using the Clinical Microbiomics Human Microbiome Profiler (CHAMP) (doi.org/10.3389/fmicb.2024.1425489).

These refinements have led to changes in our results, with only four species now showing significant associations with BP traits after adjusting for BMI. The largest drop in associated species was observed when applying the second change. We believe these updates provide a more robust and reliable interpretation of the findings.

6. Results: "We considered BMI as a potential mediator or confounder." Why not trying to address that in more detail via causal inference analysis or so? The associations 'corrupted' by BMI confounder effect are not reliable anyway.

Authors' reply: We appreciate your suggestion regarding mediation analysis. While we acknowledge that mediation analysis can technically be conducted in cross-sectional studies, we chose to refrain from performing it in this case as we do not know if BMI is a confounder or mediator. However, we have chosen to present and compare results from Model 1 (with BMI adjustment) and Model 2 (without BMI adjustment) to evaluate the effect of BMI as a covariate. The detailed results are provided in Supplementary Tables 1 and 2, and a comparison in Supplementary Figure S3. From this comparison it appears that the BMI adjustment reduces the beta estimates rather uniformly, and to a larger degree for SBP and DBP, than for SBP-SD and DBP-SD.

Supplementary Figure S3. Comparison of beta estimates in BMI-adjusted and non-adjusted models for associations with q -value < 0.05 in Model 2.

Updated text in the Discussion section:
page 24-25 lines 17-11.

The study revealed striking differences in the effect estimates between Model 1 (BMI-adjusted) and Model 2 (BMI-unadjusted) where the BMI adjustment attenuated all assessed associations. The attenuation was more prominent for 24h-SBP and 24h-DBP-phenotypes than for the SBP-SD and DBP-SD phenotypes. The substantial differences observed between BMI-adjusted and unadjusted results can be attributed to two key factors: First, BMI may serve as a proxy indicator for lifestyle factors or unmeasured confounding variables that simultaneously influence both microbial species composition and blood pressure.³⁴ Second, since certain microbial species can potentially affect host adiposity,³⁵ BMI might function as a mediator in the relationship between gut microbiota and blood pressure regulation. Given the well-established causal relationship between elevated BMI and hypertension,³⁶⁻³⁸ future longitudinal studies with comprehensive data collection are essential to elucidate whether specific gut microbiota species influence blood pressure through BMI-mediated pathways.

7. I don't get why beta diversity tests are missing. This is an important information, especially in the case of such a strong phenotype that is claimed to be associated with hundreds of species. Otherwise, why no machine learning approaches were applied to see how much of BP variance are explained by antropometrics/BMI or added by microbiome on top of that? This is again much more important information that naming of those hundreds of species and spending discussion on that.

Authors' reply: Thank you for your suggestion. The outcome of this study is blood pressure (BP), so we do not categorize BP into groups. Therefore, we did not perform beta diversity tests for group comparison. Regarding machine learning approaches to assess the contribution of variables and microbiome to BP variance: While we acknowledge the value of such analyses for prediction, the focus of our study is to emphasize the biological insights and potential mechanistic relationships between the gut microbiome and BP traits, rather than prediction accuracy. Hence, we prioritized methods that align with these biological objectives to better understand the underlying associations.

8. Conclusions: "Fourth, our findings would need replication in an independent cohort with complete 24-hour ABPM and gut microbiota data." No, the authors just showed that there's no big difference between 24hAB and office AB, so any replication cohort of appropriate size will be fine (expect for AB variability which is understudied here).

Authors' reply: Thank you for your suggestion. The updated results showed replication for only two of the four 24-hour-BP associations: *I. massiliensis* for SBP and *Dysosmobacter sp001916835* for DBP. We have revised the manuscript to reflect this finding accordingly.

Discussion

Page 25, lines 14-19

Strengths and limitations

The present study has several strengths. This is the first large population-based study to investigate the relationship of the gut microbiota with 24-hour ABPM measurements. As elaborated above, 24-hour ABPM provides a more precise and comprehensive assessment of BP compared to office measurements. The identified species-BP associations were more strongly associated with 24-hour BP compared to office BP, indicating more precise and powerful analysis using the ABPM data.

Discussion section:

page 27, lines 1-10

*Fourth, while we replicated the associations of *Intestinimonas massiliensis* with SBP and *Dysosmobacter sp001916835* with DBP using office BP in an independent subsample, several validation aspects remain important. Our findings regarding BP variability would benefit from replication in cohorts with 24-hour BP data, as this measurement cannot be captured through office visits. Future longitudinal studies with repeated microbiota sampling and BP measurements would be particularly valuable for determining whether these microbial associations are stable over time or represent transient fluctuations, and for clarifying potential causal pathways, especially regarding BMI as a possible mediator.*

9. Conclusions: “Our findings provide a starting point for further studies”. Definitely not: there many, many cohorts which studied the relationship between BP and microbiome. It’s too late for many years to claim this work as a ‘starting point’.

Authors’ reply: Thank you for your suggestion. We have revised the sentence.

Updated text in the Discussion section:
page 28, lines 3-5

Our findings on the role of human gut microbiota and its downstream metabolites in BP regulation provide potential targets for future research.

To conclude:

- More detailed attention should be given to BMI-independent associations and actual added value for 24BP

Authors’ reply: Thank you for your valuable input. We appreciate your suggestion to focus more on BMI-independent associations and the actual added value for 24-hour BP. To address this, we re-analyzed the study using Model 1 (with BMI adjustment) as our primary model and added Model 2 (without BMI adjustment) for comparison. By doing so, we aimed to better understand the contribution of microbiome-related factors to BP regulation, independent of BMI. As for the 24-hour BP, we restructure the results and discussion sections to emphasize findings related to BP variability to add value to the ABPM study.

Updated text Discussion section:
page 24, lines 4-16.

*We found that an increased gut microbiome alpha diversity was associated with lower DBP variability. Additionally, we found a member of the Oscillospiraceae family, ER4 sp900317525, to be negatively associated with SBP variability, while no species was found to be associated with DBP variability after adjusting for BMI. Oscillospiraceae are known butyrate producers and has previously been negatively associated with obesity and inflammatory bowel disease^{32 33}. Our findings add to these results proposing an inverse BMI-independent association with blood pressure variability. Since BP variability is recognized as an independent risk factor for cardiovascular events,³⁰ this finding indicates the potential importance of the gut microbiome in cardiovascular health. A previous study involving 69 participants revealed that higher levels of *Alistipes finegoldii* and *Lactobacillus* spp. were associated with lower BP variability, while *Prevotella* and *Clostridium* spp. correlated with higher variability³¹.*

Statistics approach/transformations/FDR should be revisited.

Authors’ reply: Thank you for your suggestion. In response to your concern, we have revisited both the transformation and statistical approach. We applied the centered log-ratio (CLR) transformation to address the compositional nature of microbiome data better. Additionally, we revised our statistical strategy by using two models: Model 1 (with full adjustment, including BMI) and Model 2 (without BMI adjustment). For a higher resolution of microbiome analysis, we also reanalyzed the data using the Clinical Microbiomics Human Microbiome Profiler (CHAMP), which yielded significantly fewer significant results. These changes were made to ensure a more robust analysis and improve the reliability of our findings.

Reviewer #1 (Remarks to the Author):

All my concerns have been addressed.

Authors' reply: We thank the Reviewer for the comments.

Reviewer #2 (Remarks to the Author):

1. The authors have done a good job in addressing my comments and comments of my colleague. The key points were the proper adjustment for confounders (BMI/genetics), switching focus on the added value of variable BP, and a bit more advanced composition/FDR control methods. Well done, yet there were a few points that have been missed or misinterpreted that are still worth doing: "Authors' reply: Thank you for your suggestion. The outcome of this study is blood pressure (BP), so we do not categorize BP into groups. Therefore, we did not perform beta diversity tests for group comparison." - PERMANOVA doesn't require factor or ordinal trait to correlate with dissimilarity matrix. I could be easily done on a quantitative phenotype as well. And it gives some significant messages, in particular - what's the added value pseudo-R² of (i.e. total microbiome composition variance explained) of variable BP on top of office BP? That can be done by sequential adding of terms in permanova. Adjustment of covariates can also easily be done with that.

Authors' reply: Thank you for these insightful comments. We have now used the distance-based multivariate analysis of variance (D-MANOVA) to quantify the proportion of variance in the gut microbiome composition explained by office systolic and diastolic BP, as well as mean 24-hour SBP and DBP, using the Bray-Curtis dissimilarity matrix (GUniFrac R package). First, we constructed a baseline model including age, sex, country of birth, smoking, fiber intake, total energy intake, estimated sodium intake, use of antidiabetic medication, use of antihyperlipidemic medication, BMI and technical source of variation. We then sequentially added BP variables to evaluate their incremental contribution to the microbiome composition variance. The proportion of variance explained (pseudo-R²) and statistical significance from an F-test were obtained for each model comparison. We observe that adding the 24h-BP variables added a small but significant effect.

Updated text in the Methods section, page 14, lines 11:

We used the distance-based multivariate analysis of variance (D-MANOVA) on the species Bray-Curtis dissimilarity matrix to quantify the proportion of variance explained in gut microbiome composition from blood pressure measures (GUniFrac R package). A baseline model was first constructed including age, sex, country of birth, technical source of variation, smoking, fiber intake, total energy intake, estimated sodium intake, use of antidiabetic medication, use of antihyperlipidemic medication and BMI. Office systolic and diastolic BP were then added, in separate models, to assess their contribution to microbiome variance, followed by the inclusion of the respective 24-hour BP. The proportion of variance explained (pseudo-R²) was estimated for each model

and statistical significance was assessed for each model comparison using an pseudo F-test.

Updated text in the Results section, page 18, lines 14

Contribution of office and 24-hour blood pressure to gut microbiome variance

D-MANOVA analysis showed that office BP variables explained a small but significant proportion of gut microbiome variance in addition to the baseline model (baseline pseudo-R²: 5.58%; baseline and office BP: 5.62% for SBP, $p_{\text{increment}} = 0.003$ and 5.63% for DBP, $p_{\text{increment}} < 0.001$). Adding mean 24-hour BP to these models further increased variance explained (5.67% for 24-h-SBP, $p_{\text{increment}} < 0.001$, 5.66% for 24-h-DBP, $p_{\text{increment}} < 0.020$).

Updated text in the Discussion section, page 27, lines 14

The present study has several strengths. This is the first large population-based study to investigate the relationship of the gut microbiota with 24-hour ABPM measurements. As elaborated above, 24-hour ABPM provides a more precise and comprehensive assessment of BP compared to office measurements. The identified species-BP associations were more strongly associated with 24-hour BP compared to office BP and 24-hour BP added information to the multivariate model including office BP. These findings support the superiority of ABPM data over office BP only in this context.

2. "Transformed values originally zero were replaced with the minimal non-zero transformed value for each species." - this would be a reasonable approach for metabolomics. where the detection limit might vary from one entity to another. For the sequencing it's not quite like that, and universal pseudocount might be used instead. It's also possible to deeper in the discussion on some other points raised by authors in the rebuttal, but I think these two minor aforementioned things could make manuscript great enough for the publication from both technical and scientific perspective.

Authors' reply: Thank you for pointing this out. We already used an addition of a universal pseudo-count before the clr transformation. This was unclear in our description and we have now clarified this in the text. The pseudo-count was equal to the lowest non-zero value in the relative abundance matrix. After the clr transformation, we replaced all values that were zero before transformation with the lowest value in the transformed data for each species to ensure that these values would have an equally low value after the transformation. We have updated the text in the Method section.

Updated text in the Method section, page 11, lines 19

We applied centered-log ratio (clr) transformation on the species relative abundances after an addition of a universal pseudo-value equal to the lowest non-zero abundance. After the clr transformation, those values that were originally zero were replaced with the minimal non-zero transformed value for each species. This replacement ensured that all values that were equal to zero before the transformation would have an equally low value after the transformation.

REVIEWERS' COMMENTS:

Reviewer #2 (Remarks to the Author):

my minor comments were properly taken into account. I have no more concerns. Congrats with a good paper!

Authors' reply: We thank the Reviewer for their thorough evaluation of our manuscript and for the valuable comments.